# Gluing GAP to RAS Mutants: A New Approach to an Old Problem in Cancer Drug Development

**DOI:** 10.3390/ijms25052572

**Published:** 2024-02-22

**Authors:** Ivan Ranđelović, Kinga Nyíri, Gergely Koppány, Marcell Baranyi, József Tóvári, Attila Kigyós, József Tímár, Beáta G. Vértessy, Vince Grolmusz

**Affiliations:** 1KINETO Lab Ltd., 1037 Budapest, Hungary; ivan.randelovic@kinetolab.hu (I.R.); baranyi.marcell@semmelweis.hu (M.B.); kigyos.attila@kinetolab.hu (A.K.); 2Laboratory of Genome Metabolism and Repair, Institute of Molecular Life Sciences, Research Centre for Natural Sciences, Hungarian Research Network, 1117 Budapest, Hungary; nyiri.kinga@vbk.bme.hu (K.N.); koppy.gergo@gmail.com (G.K.); 3Department of Applied Biotechnology and Food Science, BME Budapest University of Technology and Economics, 1111 Budapest, Hungary; 4Department of Pathology, Forensic and Insurance Medicine, Semmelweis University, 1091 Budapest, Hungary; jtimar@gmail.com; 5Department of Experimental Pharmacology and the National Tumor Biology Laboratory, National Institute of Oncology, 1122 Budapest, Hungary; tovari.jozsef@oncol.hu; 6Department of Computer Science, Mathematical Institute, Eötvös Loránd University, 1117 Budapest, Hungary; 7Uratim Ltd., 1118 Budapest, Hungary

**Keywords:** KRAS G12D targeting, gluing RAS mutants to GAP, pancreatic cancer therapy

## Abstract

Mutated genes may lead to cancer development in numerous tissues. While more than 600 cancer-causing genes are known today, some of the most widespread mutations are connected to the RAS gene; RAS mutations are found in approximately 25% of all human tumors. Specifically, KRAS mutations are involved in the three most lethal cancers in the U.S., namely pancreatic ductal adenocarcinoma, colorectal adenocarcinoma, and lung adenocarcinoma. These cancers are among the most difficult to treat, and they are frequently excluded from chemotherapeutic attacks as hopeless cases. The mutated KRAS proteins have specific three-dimensional conformations, which perturb functional interaction with the GAP protein on the GAP–RAS complex surface, leading to a signaling cascade and uncontrolled cell growth. Here, we describe a gluing docking method for finding small molecules that bind to both the GAP and the mutated KRAS molecules. These small molecules glue together the GAP and the mutated KRAS molecules and may serve as new cancer drugs for the most lethal, most difficult-to-treat, carcinomas. As a proof of concept, we identify two new, drug-like small molecules with the new method; these compounds specifically inhibit the growth of the PANC-1 cell line with KRAS mutation G12D in vitro and in vivo. Importantly, the two new compounds show significantly lower IC_50_ and higher specificity against the G12D KRAS mutant human pancreatic cancer cell line PANC-1, as compared to the recently described selective G12D KRAS inhibitor MRTX-1133.

## 1. Introduction

More than 600 cancer-causing mutated genes (or oncogenes) are known today [1] (cf. the Catalogue of Somatic Mutations In Cancer, https://cancer.sanger.ac.uk/cosmic, accessed on 16 February 2024); from these, numerous entries are connected to the RAS gene, whose mutations are found in approximately 25% of all human tumors [2,3]. Cancers caused by RAS mutations are some of the most difficult to treat and frequently resist chemotherapeutic attacks despite innovative novel approaches [4,5,6]. Mutations in the RAS genes, and, consequently, in the RAS proteins, are connected to the three most lethal cancers in the U.S., namely pancreatic ductal adenocarcinoma, colorectal adenocarcinoma, and lung adenocarcinoma [1]. In humans, there are three RAS isoforms: KRAS, NRAS, and HRAS; among these, the KRAS isoform is the most frequently mutated in cancers (>85%) [7]. Therefore, KRAS is one of the most important targets of drug development.

The oncogenic potential of several mutant RAS proteins is directly related to the perturbation of the RAS–GAP interaction. The wild-type RAS molecule binds the GAP (GTPase-activating protein), promotes its GTP-hydrolyzing activity, and efficiently shifts the RAS–GTP into RAS–GDP, thereby switching the RAS conformation and stopping signaling. In KRAS G12 mutants (G12C, G12D, and G12V), GAP binding cannot activate GTP hydrolysis. Hence, the signaling cascade is not terminated, and the result is an uncontrolled cell growth factor production process [8]. In order to restore normal function, the present common strategy is to shift mutant KRAS molecules into the GDP-bound (inactive) conformation. However, for a long time, the RAS protein was termed “undruggable”, since more than 30 years of drug development efforts were unsuccessful [9]. The “undruggability” of the RAS protein relates to the lack of binding cavities on the molecular surface, which is important in the oncogenic process. After many decades of fruitless efforts, two covalently bound KRAS G12C-mutant inhibitors were developed, called ARS-853 and ARS-1620, revitalizing this important research area [10,11,12,13]. Based on the success in clinical trials, Sotorasib and Adagrasib, two irreversible inhibitors, have been approved by the FDA for G12C mutant lung cancer [5,14]. Very recently, the development of noncovalent inhibitors against the KRAS G12D mutant has also been addressed, and the effects of the promising **MRTX-1133** compound in targeting G12D KRAS have been described [15,16,17], expanding the potential for the treatment of KRAS mutant tumors.

Here we present a novel strategy for finding potent new RAS inhibitors and demonstrate the potency of the method by two new molecules that inhibit human G12D KRAS mutations. Our new molecules are not covalently bound to the G12D KRAS mutation, and, consequently, they are much more drug-like than the covalently bound ARS-853 and the ARS-1620 in the case of the G12C mutation, while their activity is comparable to them.

## 2. Results

### 2.1. Identification and Characteristics of the Candidate Molecules from the Gluing Docking Strategy

The gluing strategy, with the artificially created gapped KRAS–GAP molecular structure, to which we have performed a high throughput in silico molecular docking experiment, is demonstrated in the panels of Figure 1.

Appendix A provides characteristics of the 15 candidate compounds. The structures and the docking sites of the 15 candidate compounds are shown in Figure 2.

Figure 2 shows that there are three rather separate docking pockets for the 15 compounds that involve different sites in the RAS and GAP proteins. The three separate docking pockets are shown one by one in Figure 2B–E. Compounds **2**, **3**, **8**, and **15** are grouped together in Figure 2C, while compounds **4**, **5**, **9**, **11**, **12**, and **14** are shown together in Figure 2D. Finally, Figure 2E represents a third docking site populated with compounds **1**, **6**, **7**, **10**, and **13**.

### 2.2. In Vitro Antiproliferative Activity of the Compounds and Selectivity toward KRAS^G12D^ Mutation

Table 1 shows the results of the in vitro screening of the 15 compounds on PANC-1 (KRAS^G12D^) and BxPC3 (KRAS^wt^) human pancreatic cancer cell lines after 72 h of treatment. In these experiments, compounds **10** and **14** showed the highest antitumor activity with IC_50_ values of 2.2 µM and 5.5 µM, respectively, on the PANC-1 cell line, which was significantly higher than on BxPC3. Importantly, these two compounds also showed considerable specificity ratios (selectivity) for KRAS^G12D^ mutation, with a lower IC_50_ value on the KRAS^G12D^-expressing cell line, as compared to the KRAS^wt^-expressing cell line (Table 1). Additionally, we have observed that, at lower drug concentrations, several additional compounds revealed higher antiproliferative activity against the KRAS^G12D^ cell line PANC-1 (see the IC_25_ values in Table 1). Based on the IC_25_ values, the selectivity of compounds **10** and **14** for the KRAS^G12D^ mutation cell line was also considerably higher and significant in comparison to KRAS^wt^ (Table 1).

It was of immediate interest to determine whether the newly identified compounds have similar or even better effects in the in vitro tests as compared to the recently described KRAS^G12D^ allele-specific inhibitor, **MRTX-1133** [15]. With regard to the IC_50_ values, we have observed that compounds **10** and **14** showed 8.3-fold and 3.3-fold higher antitumor activity on PANC-1, respectively, as compared to **MRTX-1133**, which was nonsignificantly and slightly more potent to this cell line in comparison to BxPC3. In addition, allele-specific selectivity for the PANC-1 KRAS^G12D^ mutation was lower for **MRTX-1133** as compared to the new compounds **10** and **14**. The same trend was observed for the IC_25_ values; in this case, however, **MRTX-1133** also showed a significant difference between the two cell lines (Table 1). To our knowledge, the effect of **MRTX-1133** on PANC-1 cells has not been investigated before. The relatively high IC_50_ value and low selectivity of **MRTX-1133** for the KRAS^G12D^ mutation in this cell line may be due to the heterozygosity of this mutation, which was earlier confirmed [18,19]. Previous studies also showed that the efficiency and selectivity of **MRTX-1133** are diversified in a large variety of cell lines harboring the KRAS^G12D^ mutation [20]. It is also worthwhile to note that, on the BxPC3 cell line, our IC_50_ data for **MRTX-1133** (20.6 µM), determined using the MTT assay, is in good agreement with the previously reported value on the same cell line (13.3 µM) [21], as determined by the CellTiter-Glo assay.

Furthermore, the two most potent and selective compounds, **10** and **14**, were also investigated on two different noncancerous cell lines in order to determine their selectivity in comparison to normal cell lines (Table 2). These data showed, with the exception of compound **10** on HUVEC-TERT cells, that both compounds are significantly less potent against the noncancerous cell lines HUVEC-TERT (umbilical vascular endothelial) and CCD-986Sk (skin fibroblast) with higher IC_50_ values compared to PANC-1, and with higher selectivity for KRAS^G12D^ mutation. Compound **14** showed lower potency against noncancerous cell lines, especially on skin fibroblast cells, suggesting that it may not be toxic for normal cells, and it is selective for cancer cells.

### 2.3. Compound 14 Shows Selective In Vivo Inhibition of Tumor Growth against KRAS^G12D^ Xenograft-Expressing Mice

Encouraged by the high antitumor potency of compounds **10** and **14** and their good selectivity toward the KRAS^G12D^ cancer cell line in comparison to the KRAS^wt^ cancer and noncancerous cell lines, we have initiated in vivo efficacy studies. However, during the estimation of the appropriate doses, compound **10** could not be dissolved for administration into the animals in a suitable dose, while compound **14** showed acceptable solubility for doses up to 6 mg/kg. Based on the solubility parameters, and the additional earlier observation that compound **14** showed better selectivity toward cancer than to noncancerous cell lines as compared to compound **10**, compound **14** was chosen for the in vivo investigation.

#### 2.3.1. Chronic Toxicity Study of Compound 14 In Vivo

In order to decide whether the dose and treatment schedule for in vivo studies are toxic or not, an in vivo chronic toxicity study with six injections of the treatment in a dose of 6 mg/kg was performed on healthy animals to determine the toxicity of compound **14**. After 15 days, the general appearance and behavior of the experimental animals were adequate, while no significant change in body weight (body weight increased in the control and treated group by 4.0 and 7.4%, respectively) (Figure 3A), as well as liver weight–body weight ratio, could be observed (Figure 3B), suggesting that compound **14** is a nontoxic substance that can be further investigated on tumor-bearing mice.

#### 2.3.2. Effect of Compound 14 in Subcutaneous Human Pancreatic PANC-1 (KRAS^G12D^) and BxPC3 (KRAS^wt^) Tumor Models In Vivo

We have shown that compound **14** has antitumor activity and selectivity toward the KRAS^G12D^ cancer cell line PANC-1 in comparison to the KRAS^wt^ cancer and noncancerous cell lines. It is also shown that the solubility of compound **14** is sufficient for further studies, and it is not toxic for animals. With these results, we progressed towards investigating the in vivo antitumor activity and specificity of compound **14** on xenograft mouse models using immunodeficient SCID male mice bearing KRAS^G12D^ (PANC-1) or KRAS^wt^ (BxPC3) human pancreatic tumors.

Based on animal body weight, which was not significantly changed in both groups and in both models during the experiment, it was shown that compound **14** is not toxic for experimental animals during the treatment at an applied dose of 6 mg/kg (Figure 4A and Figure 5A and Table 3). Moreover, the decrease in body weight was higher in the control group compared to the compound **14** treated group in both models. In addition to body weight, nonsignificant changes in the liver weight–body weight ratios confirmed the nontoxicity of this compound for experimental animals in both models (Figure 4B and Figure 5B, and Table 3).

Regarding antitumor activity, the results indicated that compound **14** reduced tumor volume, represented in mm^3^, compared to the control group in the KRAS^G12D^ mutated model of PANC-1 by 13.1% (Figure 4C and Table 3), while in the KRAS^wt^ model of BxPC3, the tumor volume was increased for 11.1% in the compound **14** treated group compared to the control (Figure 5C and Table 3), suggesting the selectivity of compound **14** in vivo toward the KRAS^G12D^ mutation.

This antitumor activity and selectivity were confirmed when evaluating the tumor volume represented by a percentage. Setting all tumor volumes as 100% at the start of treatment, and following their growth in percentage, it was observed that compound **14** inhibited tumor volume growth as compared to tumor volume in the control group for 23.8% in the PANC-1 model (Figure 4D and Table 3), while increasing tumor volume for 22.8% in BxPC3 model (Figure 5D and Table 3).

Additionally, determining the growth-rate coefficient of the tumor growth using nonlinear fitting for each tumor, it was observed that compound **14** inhibited tumor growth rate by 26.3% in comparison to the control group (*p* = 0.1348) in the KRAS^G12D^ mutated model of PANC-1 (Figure 4E and Table 3), while the growth-rate coefficient increased by the treatment in the KRAS^wt^ model of BxPC3 for 9.9% (*p* = 0.0887) compared to the control group (Figure 5E and Table 3). Moreover, the antitumor activity and selectivity toward KRAS^G12D^ mutation in vivo were confirmed by calculating the time required for tumor doubling in size. This parameter increased during the treatment with compound **14** by 14.5% (*p* = 0.1248) in the KRAS^G12D^ mutation PANC-1 bearing tumor model (Figure 4F and Table 3), while decreasing significantly by 12.1% (*p* = 0.0415) in the tumor-bearing KRAS^wt^ model (Figure 5F and Table 3), revealing that compound **14** slowed down the progression of the KRAS^G12D^ mutated tumor compared to the KRAS^wt^ tumor in vivo.

The inhibition of tumor growth by compound **14** seems to be selective against the KRAS^G12D^-expressing tumor model, as in KRAS^wt^, tumor-bearing mice inhibition of the tumor growth by the same treatment was not observed. On the contrary, treatment by compound **14** had deleterious effects on the KRAS^wt^ tumor-bearing mice. Further support for this statement is provided by comparing the growth-rate coefficient and doubling time, as independent parameters, of the compound **14** treatments in both pancreatic tumor models. In these data, we obtain a significant difference (*p* ≤ 0.0001) that compound **14** significantly decreased the growth-rate coefficient of the PANC-1 tumor, while significantly prolonging the time for tumor doubling and therefore slowing down the progression of the KRAS^G12D^ mutant tumor, compared to KRAS^wt^ BxPC3 tumor (Figure 6 and Table 3).

### 2.4. In Vitro Binding of Compound 14 to the GAP and KRAS^G12D^ Proteins

In order to investigate the direct binding of the small molecular compound **14** to GAP–KRAS^G12D^, we have applied differential scanning fluorimetry (DSF) [22]. Table 4 and Appendix A indicate that the melting points of both the GAP and KRAS^G12D^ proteins are altered in the presence of compound **14**. In addition, we have observed that, in the mixture of the GAP and KRAS^G12D^ proteins, the individual melting points detected in the solution of either GAP or KRAS^G12D^ are replaced by a single melting point, suggesting the complex formation of the two proteins. The addition of compound **14** to this mixture alters this single melting point, potentially indicating that the small molecule can also bind to the complex.

### 2.5. Molecular Interactions of Compounds 10 and 14 with the Complex of GAP and G12D Mutant KRAS

In order to investigate the detailed molecular interactions of the most efficient compounds with the complex of GAP and G12D mutant KRAS, Figure 7 and Figure 8 provide a close-up of the docking sites.

In our structural model, compound **10**, consisting of an imidazo-pyrazinyl, a thyenopyrimidine, and a phenyl group, is situated between the interaction surfaces of RAS and GAP (Figure 7). The molecule fits into the hydrophobic pocket between the C-terminal end of Switch-II and the loop between the β-5 strand and α-3 helix of RAS and into the hydrophobic gap between the α21 and α22 helices of GAP.

The pyrazole and imidazole rings of the molecule are situated in the hydrophilic pocket of KRAS formed by the main and side chains of residues E37, G60, Q61, E63, S65, and R68 that create a negatively charged protein surface patch. The NH^+^ group of the imidazole group is in an H-bond with the main chain carboxyl group of E63, while the pyrimidine ring is placed in the negatively charged pocket between the α21 and α22 helices of the GAP protein. The phenyl group of the compound faces the apolar residues of the loop between the α14 and α15 helices of GAP (residues F901, L902, L904, and I905), as well as the main chain of the 60–63 residues of KRAS.

Compound **14** is located in the KRAS–GAP interaction site near the nucleotide-binding pocket (Figure 8). The triazole and the propyl moieties of the compound are aligned between Switch-I and the turn motif between the α22 helix and β20 strand of GAP, where the triazole ring interacts with the hydrophobic side chains of Switch-I and the carbon atoms of the propyl moiety are surrounded by some apolar residues from GAP. The pyrazole and cyclohexane groups of the compound mainly interact with the GAP surface. The pyrazole ring is placed in between the guanidine groups of R789 and R894 while forming an H-bond with the T785-residue of the GAP protein, and the cycloheptane ring is placed in the pocket formed by the hydrophobic sidechains of residues at the α13 and α19 helices of GAP (L787, F788, M891, V895, and L902).

## 3. Discussion

Mutations in KRAS are associated with different types of cancer. While the G12C mutation is the most frequent in lung cancer due to smoke-induced mutation, it is very rare in the case of colorectal or pancreatic cancer. On the other hand, the G12D-type mutations are the most frequent in pancreatic and colorectal cancers [7]. Furthermore, the frequency of KRAS mutations in various cancer types is very different. In pancreatic cancer, the rate of KRAS mutation is >80%, while in lung or colorectal cancer, it is ~30%. It is another unique feature of mutant KRAS that the G12C variant in lung cancer is the major oncogenic driver, while in other tumors, mutant KRAS (which is either G12D or G12V) is a minidriver cooperating with other driver oncogenes [7,23]. All oncogenic RAS mutations affecting the G12 position manifest a loss of sensitivity to GAP activation.

In the present study, we established a new method for finding small molecules, binding to both the GAP and the mutated KRAS^G12D^ molecules, gluing them together, and thus serving as novel drug candidates for innovative cancer therapies. By this novel method in in vitro screening, we identified and selected two small molecular drugs, compounds **10** and **14**, which specifically and selectively inhibit the growth of human pancreatic cancer cell line PANC-1 with KRAS^G12D^ mutation compared to KRAS^wt^ cancer and normal cell lines with higher capacity than **MRTX-1133**.

Moreover, the inhibition of tumor growth in vivo by compound **14** under a dose of 6 mg/kg, 2–3x/week, seems to be selective against PANC-1 KRAS^G12D^ mutated pancreatic tumor model compared to BxPC3 KRAS^wt^ tumor-bearing mice, with no toxicity and side effects for the experimental animals in either model. For higher tumor inhibition and selectivity, further studies will need to increase the solubility of compound **14** to allow **the** administration of doses higher than 6 mg/kg. Alternatively, an increased frequency of administration and/or different administration routes are to be considered.

Our present findings contribute to the increasing number of compounds targeting the KRAS^G12D^ mutant. In addition to **MRTX1133** which binds preferentially to the GDP-bound KRAS^G12D^, several other compounds have already been identified for specific binding to KRAS^G12D^, and some of these entered clinical testing as well [16]. There are also several novel approaches to combine KRAS inhibitors to increase their efficacy and/or prevent immediate resistance. Two major forms are in clinical testing, the SHP2 inhibitors and the SOS1 inhibitors, which alone have not shown robust clinical activity but their combinations with RAS inhibitors may be optimal [24].

Although here we present a novel gluing approach with the identification of some novel compounds, there are also some limitations in our study that will have to be addressed in further investigations. Here, we presented in silico docking studies and, for further research, experimental determination of high-resolution three-dimensional structures would provide important insights into the mode of binding of the compounds to the RAS–GAP interface. Also, to complement the present results, the compounds could be tested further in a panel of different cell lines harboring the KRAS^G12D^ mutations or other KRAS mutations. Moreover, the solubility of compounds must be improved in order to apply a higher dose for an in vivo efficacy study and possibly to reach a better antitumor effect.

## 4. Methods and Materials

### 4.1. Identification of Small Molecules Gluing GAP and KRasG12D

The first step of the gluing approach was to generate the coordinates of a three-dimensional molecular structure consisting of the RAS and the GAP molecules, where the two proteins were close enough for the small molecules to bind to both of them (what we call “gluing”), but sufficiently apart for allowing in the small molecules. This approach would establish novel binding sites for small, drug-like molecules between the “undruggable” RAS and the GAP, which the RAS structure, by itself, is lacking.

Our present approach suggests docking to the suitably configured RAS-GAP system, where the “suitable configuration” covers the creation of a theoretical (meaning that it is not measured by crystallography) configuration of RAS and GAP. The original configuration of the RAS-GAP complex was the wild-type 1WQ1 PDB entry [25]. First, we in silico mutated residue 12 (glycine) in RAS (chain ID R) to aspartate in the structure 1WQ1, using the built-in software of PyMol (version 0.99). Next, we proceeded to create an artificially altered (theoretical) complex structure where the RAS and GAP proteins are pulled apart from each other to generate a space between the two protein molecules. This artificial distancing of the two proteins was achieved by moving the GAP protein away from RAS. To achieve this, we applied a translation on all of the atoms of the GAP protein using a 5 Å long *v* vector (Figure 1). The length and the direction of the *v* translational vector was chosen such as to maximize the possible binding surfaces around the active site and giving space for the gluing molecules between RAS and GAP proteins. The direction of the translational vector *v* was selected to be approximately parallel to the normal vector of the plane, which divides the RAS and GAP molecules around the active site, and the length was selected to be 5 Angström, which is 3–4 times larger than the length of a covalent bond in a residue. As a result of this transformation, an “artificial” configuration was created, where there is a suitable place for small, drug-like molecules to bind to both RAS and GAP; therefore, gluing them even in the mutated state.

The artificially created GAP-RAS configuration represented on Figure 1, panel B, served as a receptor for the high-throughput in silico molecular docking software FRIGATE (version 1.00) [26]. This was used for docking 4.6 million small, drug-like molecules from the “clean leads” subset of ZINC 12 database [27]. Each of the 4.6 million drug-like small molecules were processed with the same method. All rotational bonds of each small molecule were allowed to rotate in the docking procedure, which was done by the FRIGATE docking tool to minimize the joint energy of the protein-ligand complex. The starting configurations of the drug molecules were identified in the ZINC database. Using this starting configuration, for a molecule with k rotational bonds, a point in the 3 + 3 + k dimensional Euclidean space represented the configuration of the drug molecule: 3 + 3 dimensions for representing the molecule as a rigid structure (3 spatial dimensions plus the 3 Eulerian angles); and one additional dimension for each rotating bond. The FRIGATE docking tool minimizes the joint energy of the receptor-ligand system by combining a discrete and a continuous approach: before starting the docking procedure of millions of small molecules, it computes the potential of each possible ligand-atom in a dense rectangular grid around and in the receptor (it is the discrete step); next, by using spline-approximation from the grid points, it uses a gradient-based local optimization (it is the continuous step), enhanced by a heuristic global optimization strategy. The discrete step makes the optimization computationally feasible; the continuous step makes it efficient in finding local minima. The details of the FRIGATE tool are described in [26]. The most favorable small molecular hits are identified and listed in the molecular docking process. Fifteen of the best-scored molecules were acquired from vendors (cf. Appendix A), and their anti-cancer activity was investigated in human cancer cell line cultures. 

### 4.2. Cell Lines and Culture Conditions

In experimental procedures, we used pancreatic cancer cell lines PANC-1 (KRAS^G12D^) and BxPC3 (KRAS^wt^), which were cultured in RPMI 1640 Medium with glutamine (Roswell Park Memorial Institute Medium, Biosera, Nuaille, France), while non-cancerous cell lines CCD-986Sk (skin fibroblast) and HUVEC-TERT (umbilical vascular endothelial) were cultured in Iscove’s Modified Dulbecco’s Medium (Biosera) and in Endothelial cell basal medium-2 (EBM-2; Lonza, Basel, Switzerland), respectively. RPMI and IMDM mediums were supplemented with 10% heat-inactivated Fetal Bovine Serum (FBS; Biosera) and 1% Penicillin/Streptomycin (Biosera), while CCD-986Sk cells were cultured in 20% FBS-containing medium. Medium for HUVEC-TERT cells was supplemented with the Endothelial Growth Medium-2 kit (EGM-2; Lonza) by manufacturer instructions. The cell lines were obtained from the American Type Culture Collection (ATCC). Cells were cultured in sterile T25 or T75 flasks with ventilation caps (Sarstedt, Nümbrecht, Germany) at 37 °C in a humidified atmosphere with 5% CO_2_.

### 4.3. In Vitro Antiproliferative Activity of Compounds and Calculation of Selectivity for KRAS^G12D^ Mutation

For the evaluation of the in vitro antiproliferative activity of compounds, the cell viability was determined by MTT assay (3-(4,5-dimethylthiazol-2-yl)-2,5-diphenyl-tetrazolium bromide) which was obtained from Duchefa Biochemie (Haarlem, The Netherlands). After standard harvesting of the cells by trypsin-EDTA (Biosera) and phosphate-buffered saline (PBS; Biosera), 7 × 10^3^ cells per well for pancreatic cancer cell lines and 10 × 10^3^ cells per well for non-cancerous cell lines were seeded in 5% FBS-containing growth medium to 96-well plates with flat bottom (Sarstedt), in a 100 μL volume per well, and incubated at 37 °C. After 24 h, cells were treated with various concentrations of compounds (15 nM–100 µM), dissolved in dimethylsulfoxide (DMSO; Sigma Aldrich, St. Louis, MO, USA, 0.5% final) and FBS-free medium (FBS final 2.5%) and incubated for 72 h under standard conditions. The control wells were treated with FBS free medium (FBS final 2.5%) and 0.5% DMSO final. Afterward, the MTT assay was performed in order to determine cell viability, by adding 22 µL of MTT solution (5 mg/mL in PBS, 0.5 mg/mL final) to each well and after 2 h of incubation at 37 °C, the supernatant was removed. The precipitated purple formazan crystals were dissolved in 100 µL of a 1:1 solution of DMSO—96% Ethanol (Molar Chemicals Kft., Halásztelek, Hungary), and the absorbance was measured after 15 min. at λ = 570 nm by using CLARIOstar^plus^ microplate reader (BMG Labtech, Ortenberg, Germany). Average background absorbance (DMSO-Ethanol) was subtracted from absorbance values of control and treated wells, and cell viability was determined relative to untreated (control) wells where cell viability was arbitrarily set to 100%. Absorbance values of treated samples were normalized versus untreated control samples and interpolated by non-linear regression analysis with GraphPad Prism 6 software (GraphPad, La Jolla, San Diego, CA, USA) to generate sigmoidal dose-response curves from which the 50% inhibitory concentration (IC_50_) values of the compounds were calculated and presented as micromolar (µM) units. The experiments were done in triplicate, and each experiment was repeated three times. Selectivity of the compounds toward KRAS^G12D^ mutation compared to KRAS^wt^ is calculated based on the ratio between IC_50_ of KRAS^wt^ or non-cancerous cell line and IC_50_ of KRAS^G12D^ mutated cell line. Selectivity values higher than 1 represent selectivity toward KRAS^G12D^ mutation compared to KRAS^wt^, while values lower than 1 represent the opposite.

### 4.4. Experimental Animals

Adult female BALB/c mice were used in the chronic toxicity study, while the immunodeficient SCID male mice were used in subcutaneous PANC-1 and BxPC3 human pancreatic tumor models in vivo experiments. Mice were held in filter-top boxes in the experimental barrier rooms, and every box opening was performed under a laminar-flow hood in sterile conditions. The cage components, corn-cob bedding, and food (VRF1 from Special Diet Services) were steam-sterilized in an autoclave (121 °C, 20 min). The distilled water was acidified to pH 3 with hydrochloric acid. The animals used in these studies were cared for according to the “Guiding Principles for the Care and Use of Animals” based upon the Helsinki Declaration, and they were approved by the local ethical committee. The animal housing density was according to regulations and recommendations from directive 2010/63/EU of the European Parliament and of the Council of the European Union on the protection of animals used for scientific purposes. Permission license for breeding and performing experiments with laboratory animals: PEI/001/1715-2/2015 and PE/EA/401-7/2020.

### 4.5. Chronic Toxicity Study of Compound **14**

In order to determine the toxicity of compound **14** on healthy animals, a chronic toxicity study was performed. Adult BALB/c female mice (18–20 g), which were kept under the conditions as described above, were treated with compound **14** by intraperitoneal (i.p.) administration with a dose of 6 mg/kg in a volume 0.1 mL per 10 g of mice body weight, on days 1, 3, 5, 8, 10 and 12. In the case of the control group, 8% Ethanol (Molar Chemicals Kft.)/8% DMSO (Sigma Aldrich) in sterile water for injection (Pharmamagist Kft., Budapest, Hungary) as solvent was administered. Each group consisted of three mice. The toxicity was evaluated on the basis of life span, liver toxicity, behavior and appearance of the mice, as well as the body weight. Parameters were followed for 15 days.

### 4.6. Mouse Models of Subcutaneous Human Pancreatic Cancers PANC-1 (KRAS^G12D^) and BxPC3 (KRAS^wt^), Doses and Schedule of Compound **14** Treatments, and Measurements

Adult SCID male mice (32–41 g), 38 of them, were used in this experiment and kept under the conditions described above. PANC-1 (KRAS^G12D^) and BxPC3 (KRAS^G12D^) human pancreatic cancer cells were injected subcutaneously into mice, whereby 3 × 10^6^ cells were used per animal, suspended in 200 µL of RPMI medium (Biosera). Treatments started 23 and 7 days after cells inoculation, when average tumor volume was 64.0 and 55.4 mm^3^, respectively, for PANC-1 and BxPC3 tumor- bearing mice. Compounds were dissolved in 8% Ethanol (Molar Chemicals Kft.) and 8% DMSO (Sigma Aldrich) in sterile water for injection (Pharmamagist Kft.) solution and administered by i.p. injection in a volume 0.1 mL per 10 g of mice body weight, 3 times per week. For the PANC-1 model 2 groups with 9 animals per group, while in BxPC3 model 2 groups with 10 animals per group were established. The mice in the control group were treated with the solvent. Animals bearing PANC-1 tumor were treated by the next schedule and doses: with 6 mg/kg in 8% Ethanol/8% DMSO/water on days 24, 27, 29, 31, 34, 36, 38, 41, 43, 45, 48, 50, 55, 57, 59, 62, 64, 66, 69, 71, 73, 76 and 78 after cell inoculation. Animals bearing BxPC3 tumor were treated by the next schedule and doses: with 6 mg/kg in 8% Ethanol/8% DMSO/water on days 8, 11, 13, 15, 18, 21, 25, 28, 32, 35, 39, 41, 43, 46 and 48 after cell inoculation. Tumor volumes were measured initially when the treatment started and at periodic intervals. A digital caliper was used to measure the longest (a) and the shortest diameter (b) of a given tumor. The tumor volume was calculated using the formula V = ab^2^ × π/6, whereby a and b represent the measured parameters (length and width). The termination of the experiment was 80 days after cell inoculation, i.e., 57 days after treatment started for the PANC-1 model, and 53 days after cell inoculation, i.e., 46 days after treatment started for the BxPC3 model, since the average volume of the tumors in the control (PANC-1) and in compound **14** (BxPC3) groups reached over 1600 mm^3^. The mice from all groups were sacrificed by cervical dislocation, after which their tumors and livers were harvested.

The antitumor effect of compound **14** was evaluated by measuring the tumor volume and calculating the percentage of how much tumor volume grew in comparison to the starting tumor volume, which was set for all tumors as 100% at the start of treatment. Additionally, the growth-rate coefficient of the tumor growth was determined using non-linear fitting for each tumor and average was calculated for each group. The doubling time of the tumor was calculated similarly. The toxicity effects of the compound were evaluated by measuring the animal body and liver weights and calculating the liver weight/body weight ratio.

### 4.7. Protein Expression and Purification

KRAS G12D (residues 1–169) was expressed with the same method applied for KRAS-G12C in our previous work [28].

The gene encoding the catalytic domain of GAP (714-1047) residues Uniprot ID: P20936) was ordered from GeneScript in a Pet-22 plasmid. The protein was expressed in *E. coli* BL21 Rosetta cells at 18 °C overnight after induction with 0.5 mM IPTG at OD600 = 0.6. The cells were lysed in 50 mM TRIS, 500 mM NaCl, 5 mM imidazole, pH 8.0 and purified on a Ni NTA column. After elution with 250 mM imidazole the protein was dialyzed to 20 mM HEPES, 150 mM NaCl, pH 7.5 buffer.

The purity of protein preparations was >95% based on SDS PAGE analysis.

### 4.8. Differential Scanning Fluorimetry

Samples containing either GTP loaded KRAS-G12D or GAP or both proteins at 20 μM with or without compound **14** at 600 μM concentration and SYPRO^®^ Orange 5X (1000-fold dilution) were heated from 25 to 85 °C with the speed of 1.5 °C/min in a BioRad CFX96 Touch instrument (Hercules, CA, USA) and fluorescence on HEX channel was detected. Each sample was 25 μL, contained 5% DMSO. Samples were measured in triplicates. Melting points were determined as the minimum value of the first negative derivate of the obtained protein melting curves.

### 4.9. Statistical ANALYSIS

In vitro data are shown as mean ± standard deviation (SD), and in vivo data are presented as mean ± standard error of the mean (SEM). Statistical analyses were performed by GraphPad Prism 6 (GraphPad Software, San Diego, CA, USA) using two-tailed unpaired *t*-test with Welch’s correction to analyse statistical differences between groups. The experimental data where *p*-values lower or equal than 0.05 were considered statistically significant. The symbols *, **, *** and **** refer to significance at *p* ≤ 0.05, *p* ≤ 0.01, *p* ≤ 0.001 and *p* ≤ 0.0001, respectively.

## 5. Conclusions

Our present results indicate that the large scale and high throughput docking may be useful when paired with the novel gluing strategy to identify promising compounds against mutant KRAS. It is of great importance to note that the binding mode of the previously suggested G12D-specific **MRTX-1133** compound is associated with different characteristics compared to the binding observed in the structural model of our new compounds **10** and **14** (Figure 9). This observation suggests that further studies for inhibitor design may also focus on an expanded surface of the GAP–RAS complex.

## Figures and Tables

**Figure 1 ijms-25-02572-f001:**
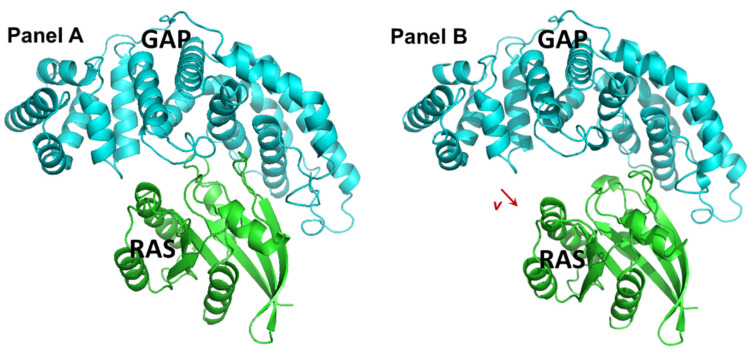
Construction of the receptor complex to be used in the gluing docking. Panel (**A**) visualizes the 1WQ1 complex structure from the PDB; proteins RAS (green) and GAP (cyan) are in the cartoon model. Panel (**B**) shows the result of the translational movement where the GAP protein molecule was translated by a 5-Å-long *v* vector (*v* = (−4.903, −0.799, 0.567), shown as the red arrow on Panel **B**).

**Figure 2 ijms-25-02572-f002:**
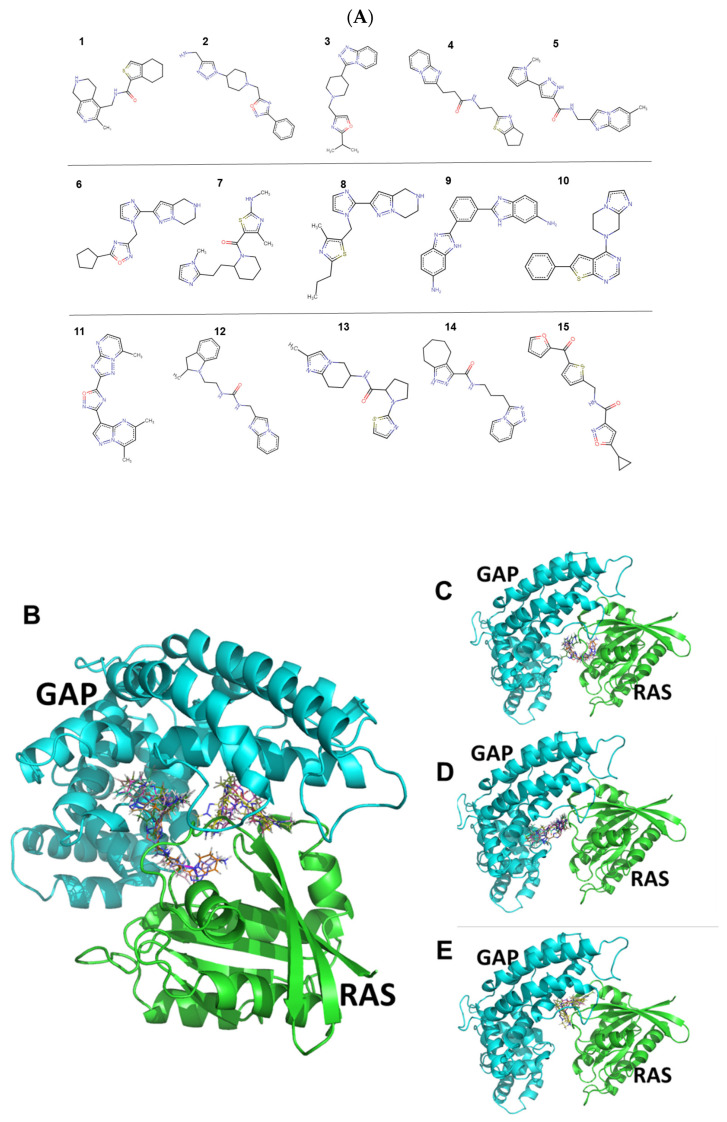
Structural formulae (Panel (**A**)) and docking conformations (Panels (**B**–**E**)) of the 15 candidate compounds, as identified during the gluing docking method. Proteins RAS (green) and GAP (cyan) are shown in the cartoon model, while docked compounds are shown in lines. The figure was created by Pymol (the Pymol file is provided in the Supplementary Data Set S1).

**Figure 3 ijms-25-02572-f003:**
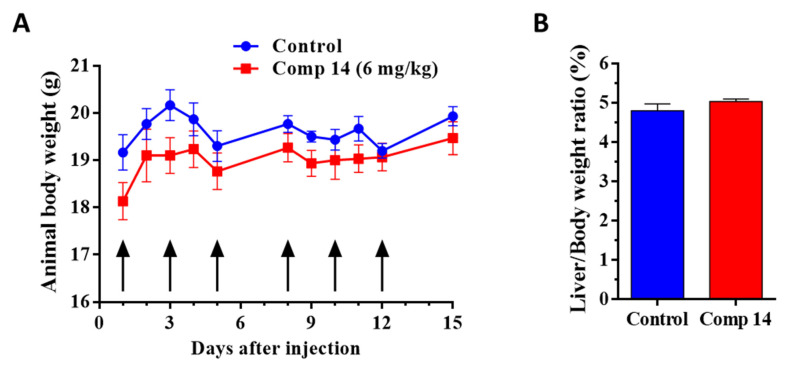
In vivo chronic toxicity study of compound **14** on healthy BALB/c female mice with 6 treatments (black arrows) under a dose of 6 mg/kg administered intraperitoneally. (**A**) Animal body weight (grams, average ± SEM). (**B**) Liver weight–body weight ratio (percentage, average ± SEM) after the termination of the experiment, 15 days subsequent to the first treatment. Three animals were used per group. Statistical analysis was performed by a two-tailed unpaired *t*-test with Welch’s correction. Nonsignificant differences are not marked.

**Figure 4 ijms-25-02572-f004:**
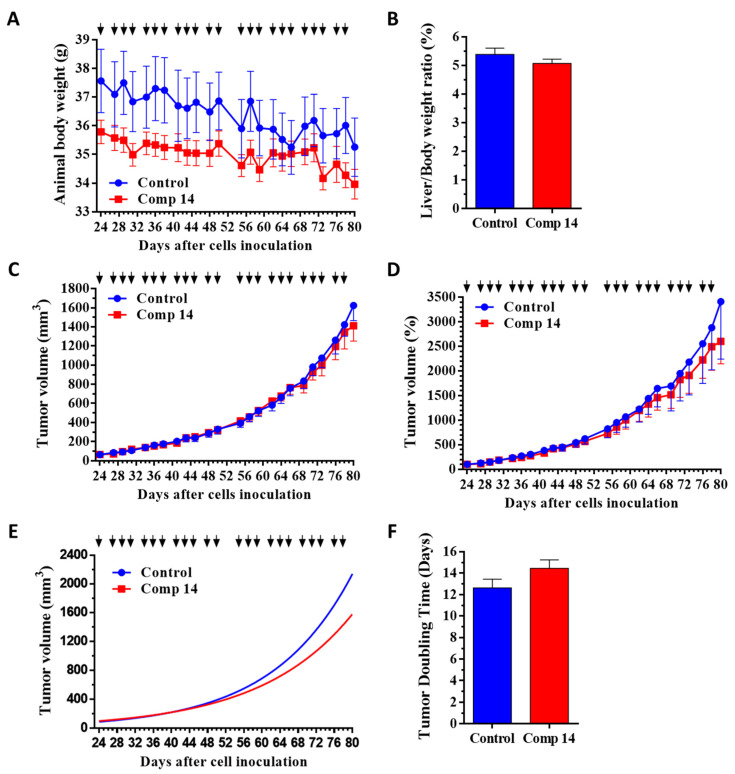
Effect of compound **14** (6 mg/kg, black arrows, administered intraperitoneally) in subcutaneous human pancreatic PANC-1 (KRAS^G12D^) tumors bearing SCID male mice. (**A**) Animal body weight (grams, average ± SEM). (**B**) Liver weight–body weight ratio (percentage, average ± SEM) after termination of the experiment, 80 days after cell inoculation. (**C**) Tumor volume (mm^3^, average ± SEM). (**D**) Tumor volume (percentage, average ± SEM). (**E**) Tumor volume by nonlinear fitting (mm^3^, average). (**F**) Tumor doubling time (days, average ± SEM). Nine animals were used per group. Statistical analysis was performed by a two-tailed unpaired *t*-test with Welch’s correction. Nonsignificant differences are not marked.

**Figure 5 ijms-25-02572-f005:**
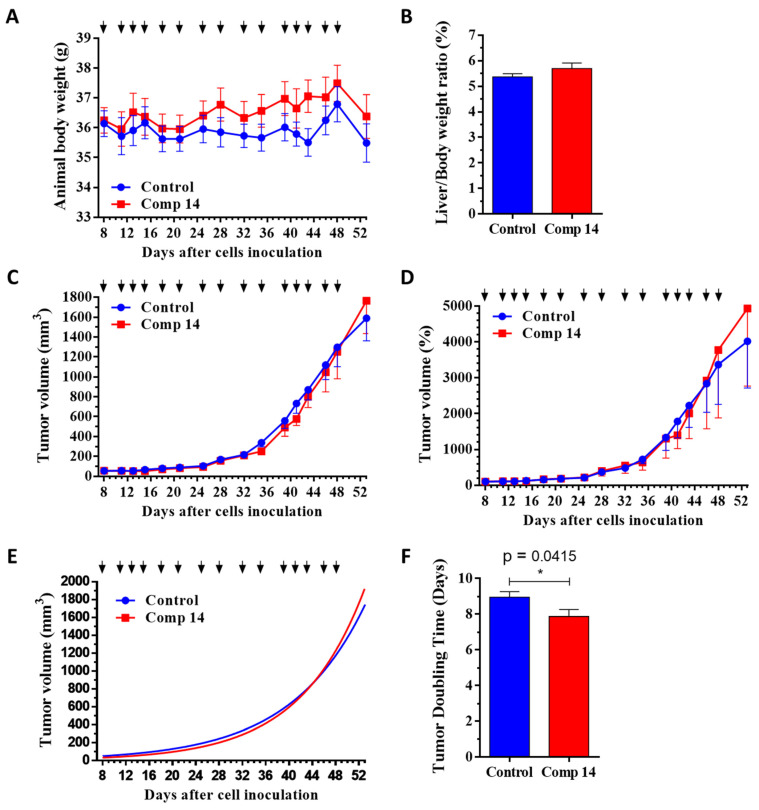
Effect of compound **14** (6 mg/kg, black arrows, administered intraperitoneally) in subcutaneous human pancreatic BxPC3 (KRAS^wt^) tumors bearing SCID male mice. (**A**) Animal body weight (grams, average ± SEM). (**B**) Liver weight–body weight ratio (percentage, average ± SEM) after the termination of the experiment, 53 days after cell inoculation. (**C**) Tumor volume (mm^3^, average ± SEM). (**D**) Tumor volume (percentage, average ± SEM). (**E**) Tumor volume by nonlinear fitting (mm^3^, average). (**F**) Tumor doubling time (days, average ± SEM). Ten animals were used per group. Statistical analysis was performed by a two-tailed unpaired *t*-test with Welch’s correction. Nonsignificant differences are not marked. The symbol blue * means a significant difference compared to the control group at *p* ≤ 0.05.

**Figure 6 ijms-25-02572-f006:**
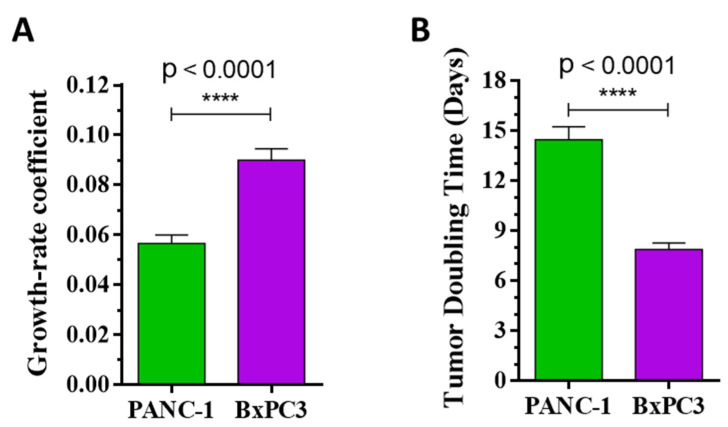
Comparison effect of compound **14** (6 mg/kg, administered intraperitoneally) in subcutaneous human pancreatic PANC-1 (KRAS^G12D^, green) and BxPC3 (KRAS^wt^, violet) mice models. (**A**) Tumor growth-rate coefficient (value, average ± SEM). (**B**) Tumor doubling time (days, average ± SEM). Statistical analysis was performed by a two-tailed unpaired *t*-test with Welch’s correction. The symbol **** means significant difference compared to the control group at *p* ≤ 0.0001.

**Figure 7 ijms-25-02572-f007:**
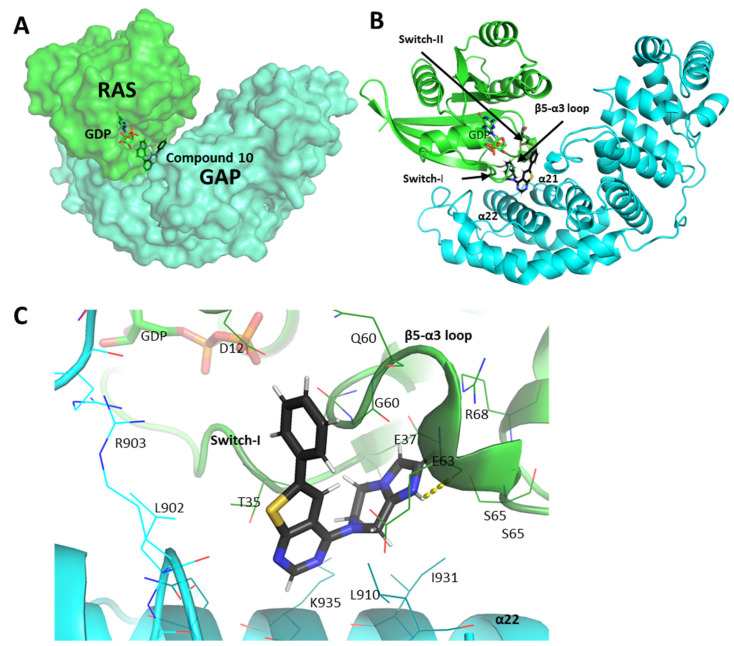
Docking site of compound **10** in the structural model of KRAS-G12D and GAP. (**A**) shows compound **14** upon the protein surface, while (**B**) shows the compound with the secondary/tertiary structural elements of the proteins. (**C**) is a close-up image of the binding site, with the nearby potentially interacting residues shown as sticks. RAS is shown in green; GAP is shown in cyan. Darker shades on the cartoons and sticks indicate residues within 4 Å of the compound. Legends in bold indicate secondary structural elements or domains of the proteins, while residues are indicated by light legends. H-bonds are shown as yellow dashed lines. Figures were created by the PyMol visualization tool. The RAS–GAP configuration and the molecule docking process were done as described in the Section 4.

**Figure 8 ijms-25-02572-f008:**
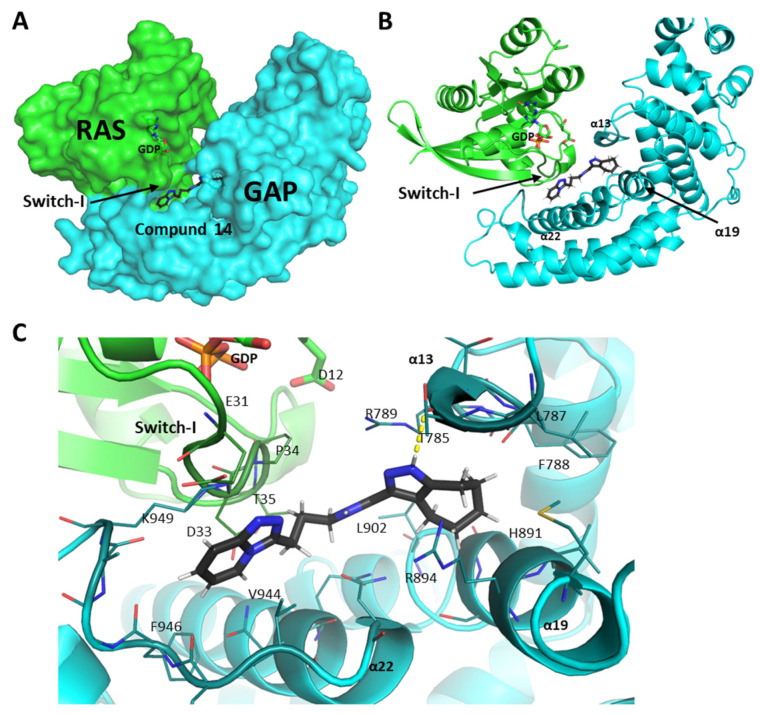
Docking site of compound **14** in the structural model of KRAS-G12D and GAP. (**A**) shows molecule-14 in relation to the protein surface, while (**B**) shows the compound in relation to the secondary structural elements of the proteins. (**C**) is a close-up image of the binding site, with the nearby, potentially interacting residues shown as sticks. KRAS is shown as green; GAP is shown as cyan. Darker shades on the cartoons and sticks indicate residues within 4 Å of the compound. Legends in bold indicate secondary structural elements, or domains of the proteins, while residues are indicated by light legends. Figures were created by the PyMol visualization tool. The KRAS–GAP configuration and the molecule docking process were done as described in the Section 4.

**Figure 9 ijms-25-02572-f009:**
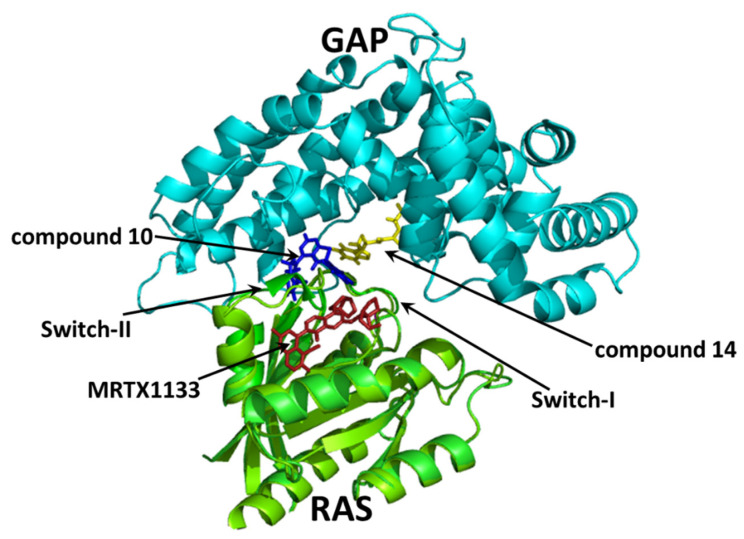
Binding of MRTX-1133 compared to the docking sites of compounds **10** and **14**. The RAS^G12D^–**MRTX1133** complex (PDB ID: 7RPZ [15]) (RAS protein in light green cartoon model, **MRTX1133** molecule in red sticks) is superimposed upon the receptor RAS^G12D^–GAP complex (created as described in the Methods). In the receptor RAS^G12D^–GAP complex, the docking positions of the newly identified compounds **10** (blue sticks) and **14** (yellow sticks) are also shown. GAP is in cyan in the cartoon model; RAS is green in the cartoon model. The positions of the RAS switch-I and RAS switch-II regions are indicated with arrows.

**Table 1 ijms-25-02572-t001:** Antiproliferative effect of 15 compounds and **MRTX-1133**, after 72 h treatment, on PANC-1 (KRAS^G12D^) and BxPC3 (KRAS^wt^) human pancreatic cancer cell lines and their selectivity toward KRAS^G12D^ mutation. IC_50_ and IC_25_ values in average ± SD are presented only for compounds **10**, **14**, and **MRTX-1133** (the experiments were done in triplicate and each experiment was repeated three times). The other compounds showed very low potency and low selectivity toward KRAS^G12D^ mutation in the first screening. Hence, these were not further tested. Selectivity represents the ratio between IC_50_BxPC3^KRAS-wt^ and IC_50_PANC-1^KRAS-G12D^ values. Selectivity values higher than 1 represent selectivity toward KRAS^G12D^ mutation compared to KRAS^wt^, while values lower than 1 represent the opposite.

Compound Name	IC_50_ (µM)		Selectivity toward KRAS^G12D^ Mutation	IC_25_ (µM)		Selectivity toward KRAS^G12D^ Mutation
	PANC-1 (KRAS^G12D^)	BxPC3 (KRAS^wt^)		PANC-1 (KRAS^G12D^)	BxPC3 (KRAS^wt^)	
**1**	53.77	48.45	0.9	17.36	30.99	1.8
**2**	69.13	35.05	0.5	16.09	19.35	1.2
**3**	95.09	42.30	0.4	31.14	24.84	0.8
**4**	93.69	39.91	0.4	28.40	22.67	0.8
**5**	118.10	30.76	0.3	32.84	14.85	0.5
**6**	93.52	45.08	0.5	41.55	27.30	0.7
**7**	138.60	40.55	0.3	52.20	24.16	0.5
**8**	90.71	43.73	0.5	26.84	27.07	1.0
**9**	74.12	23.23	0.3	31.14	13.44	0.4
**10**	2.2 ± 0.4 *	3.8 ± 0.1	1.7	0.05 ± 0.01 ##	0.5 ± 0.05	10.6
**11**	117.30	43.18	0.4	50.28	25.30	0.5
**12**	93.11	42.97	0.5	31.41	25.12	0.8
**13**	103.20	47.54	0.5	41.12	29.96	0.7
**14**	5.5 ± 1.3 *	8.2 ± 0.6	1.5	1.2 ± 0.4 ###	4.5 ± 0.4	3.8
**15**	78.40	55.75	0.7	20.72	35.15	1.7
**MRTX-1133**	18.3 ± 2.4 ^ns^	20.6 ± 4.0	1.1	5.7 ± 2.4 #	12.3 ± 2.7	2.2

Two-tailed unpaired *t*-test with Welch’s correction was used to analyze statistical differences between IC_50_ and IC_25_ values of compounds **10**, **14**, and **MRTX-1133** on two cell lines. The symbols * and # mean that compound is significantly more potent (lower IC_50_ and IC_25_, respectively) and with higher selectivity for KRAS^G12D^ on PANC-1 (KRAS^G12D^) compared to BxPC3 (KRAS^wt^). The symbols *, #, ## and ### mean significant at *p* ≤ 0.05, *p* ≤ 0.01 and *p* ≤ 0.001, respectively. ns means nonsignificant difference.

**Table 2 ijms-25-02572-t002:** Antiproliferative effect of compounds **10** and **14**, after 72 h treatment, on noncancerous cell lines HUVEC-TERT (umbilical vascular endothelial) and CCD-986Sk (skin fibroblast) and their selectivity toward KRAS^G12D^ mutation in comparison to noncancerous cells. IC_50_ values are shown as (average ± SD). Selectivity represents the ratio of the IC_50_ values determined for noncancerous cell lines and the PANC-1^KRAS-G12D^ cell lines. Selectivity values higher than 1 represent selectivity toward KRAS^G12D^ mutation compared to noncancerous cells, while values lower than 1 represent the opposite. The experiments were done in triplicate and each experiment was repeated three times.

Noncancerous Cell Line	Compound 10		Compound 14	
	IC_50_ (µM)	Selectivity toward KRAS^G12D^ Mutation	IC_50_ (µM)	Selectivity toward KRAS^G12D^ Mutation
HUVEC-TERT	0.3 ± 0.04 ^+^	0.1	8.6 ± 0.7 *	1.6
CCD-986Sk	6.7 ± 1.1 *	3.0	25.6 ± 3.0 **	4.7

Two-tailed unpaired *t*-test with Welch’s correction was used to analyze statistical differences between IC_50_ values of compounds **10** and **14** on PANC-1 (KRAS^G12D^) from Table 1 compared to noncancerous cell lines. The symbols * and ** mean that the compound is significantly more potent (lower IC_50_) and with higher selectivity for KRAS^G12D^ on PANC-1 (KRAS^G12D^) compared to noncancerous cell lines at *p* ≤ 0.05 and *p* ≤ 0.01, respectively. The symbol ^+^ means significant at *p* ≤ 0.05 but with selectivity for the noncancerous cell line, not for KRAS^G12D^.

**Table 3 ijms-25-02572-t003:** Effect of compound **14** administered intraperitoneally in subcutaneous human pancreatic PANC-1 (KRAS^G12D^) and BxPC3 (KRAS^wt^) tumors bearing mice. Data represent percentage values (%) where negative values indicate a decrease of the animal body weight at the end of the experiment, compared to the start; a decrease of the liver weight–body weight ratio compared to the control group; inhibition of tumor growth compared to the control group; and decrease of the time for tumor doubling compared to the control group. Statistical analysis was performed by a two-tailed unpaired *t*-test with Welch’s correction. Nonsignificant differences are not marked. The symbols * and **** mean significant difference compared to the control and compound **14** treated groups of BxPC3 at *p* ≤ 0.05 and *p* ≤ 0.0001.

Parameter	Control		Comp 14	
	PANC-1	BxPC3	PANC-1	BxPC3
Animal body weight	−6.1	−1.8	−5.1	+0.4
Liver/Body weight ratio			−5.8	+5.8
Tumor volume in mm^3^			−13.1	+11.1
Tumor volume in %			−23.8	+22.8
Tumor volume by nonlinear fitting			−26.3 (**** to BxPC3)	+9.9
Tumor doubling time			+14.5 (**** to BxPC3)	−12.1 (* to control)

**Table 4 ijms-25-02572-t004:** Binding of compound **14** to GAP and KRAS^G12D^ proteins followed by DSF.

	Melting Point, °C
GAP	42.5 ± 0.20
GAP + compound **14**	40.5 ± 0.15
KRAS^G12D^	55.5 ± 0.30
KRAS^G12D^ + compound **14**	53.0 ± 0.35
GAP + KRAS^G12D^	43.5 ± 0.10
GAP + KRAS^G12D^ + compound **14**	40.5 ± 0.25

Table indicates average data and SD from three independent experiments, run in triplicates.

## Data Availability

All data generated during this study are included in this article.

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
