# Peer review of "Gluing GAP to RAS Mutants: A New Approach to an Old Problem in Cancer Drug Development"

_ijms, 2024, doi:10.3390/ijms25052572_

Round 1
Reviewer 1 Report
Comments and Suggestions for Authors
Authors have presented a novel method to design drugs against KRAS protein which was considered an undruggable drug target for several years. They show that by using docking they were able to identify KRAS inhibitors that glue KRAS with GAP. They also showed that identified small molecules had effect on G12D and WT KRAS cell lines and mice. Although this work is novel and very important knowing the fact that RAS is responsible for almost 30 percent of all tumors, there are a few concerns that authors should address.
1) The binding affinity and specificity of MRTX1133 are well established. The authors did not discuss how their data for MRTX1133 compares with published data. Based on the reported data MRTX1133 is not very specific to G12D which is not the case. The authors should explain any discrepancies.
2) How many biological and experimental replicates were used is missing. Also, statistical analysis should be elaborated
3) it is not mentioned how the docking model of compounds 10 and 14 was generated and whether it was validated with more than one docking method.
4) There are a few pockets (eg Switch II pocket, Switch II/III pocket) that have been identified in KRAS. The authors did not mention if compound 10 or 14 binds to any of the known pockets or a new pocket. If authors identified a new pocket they show that pocket in KRAS structure.
5) One of the major drawbacks of the current work is, that the authors did not perform any assay to show that identified small molecules directly bind to KRAS or inhibit KRAS-mediated pathways.
6) Most of the figures with structures lack labels like what protein and where are switch I and II etc. Please add them.
Author Response
Enclosed in pdf.

Reviewer 2 Report
Comments and Suggestions for Authors
The manuscript: “Gluing GAP to RAS Mutants: A New Approach to an Old Problem in Cancer Drug Development” represents an interesting original scientific paper describing the impact of new compounds that specifically inhibit the growth of PANC-1 cell line with KRAS mutation G12D in vitro and in vivo.
The entire manuscript is well written with a sufficiently explained Materials and Methods section and an adequately described, well-illustrated, and tabular presented result section. The presented results support the following conclusion, and a corresponding reference list accompanies the manuscript.
The acceptance of the manuscript in its current form is suggested.
Author Response
Enclosed in pdf.

Reviewer 3 Report
Comments and Suggestions for Authors
The study focuses on the development of novel inhibitors targeting KRAS mutations, which are implicated in various lethal cancers​​. The use of both in vitro and in vivo models, including cell lines and animal studies, to evaluate the efficacy and toxicity of the compounds is commendable​​. The manuscript presents an intriguing new approach to targeting KRAS mutations in cancer, with promising preliminary results. The study's innovative methodology, combined with comprehensive in vitro and in vivo analyses, makes a valuable contribution to the field. However, enhancement in methodological details, statistical analysis, and discussion of broader implications and limitations would strengthen the manuscript. Major revision is recommended.
1.While the overall experimental approach is innovative, the manuscript could benefit from more detailed methodological descriptions, particularly regarding the computational aspects of the "gluing" strategy.
2.The manuscript should include more detailed statistical analysis, especially in the in vitro and in vivo study sections. This would help in understanding the significance of the observed results more clearly.
3.Broader Implications and Future Directions: The discussion could be expanded to include the broader implications of these findings in the field of cancer therapeutics and potential future directions for this line of research.
4.Authors should discuss the limitations of their study more thoroughly, including any potential challenges in translating these findings from a laboratory setting to clinical application.
5. The type of this article should be reconsidered.
Author Response
Enclosed in pdf.

Round 2
Reviewer 1 Report
Comments and Suggestions for Authors
The authors have incorporated all suggested changes and the revised manuscript is in good shape. Only small changes are needed. In Figure 9, switches should be labeled. Also, Ras in Figure 9A and 9B should be in the same orientation or overlaid with each other for a fair comparison of binding of MRTX1133 and Compound 10/14.
Author Response
Dear Reviewer,
Thank you for your review on our manuscript and for accepting our changes following your suggestions. In order to address your comment on our revised manuscript:
“In Figure 9, switches should be labeled. Also, Ras in Figure 9A and 9B should be in the same orientation or overlaid with each other for a fair comparison of binding of MRTX1133 and Compound 10/14.”
In the revised version of our manuscript, we have provided a new Figure 9 where we provide an overlay with the RAS:MRTX1133 complex and our presently introduced RAS:GAP complex together with the docking positions for compounds 10 and 14. We fully agree that in this overlay, the relative positions of MRTX1133 and compounds 10 and 14 are much better shown. We have also labelled the switch positions.
Thank you for your suggestion.
Reviewer 3 Report
Comments and Suggestions for Authors
Thank the authors for their revision, acceptance is recommended.
Author Response
Dear Reviewer,
we thank you for your review and for accepting our revised version. We are grateful for the comments in your first review that helped to improve our study.
Best regards
Vince Grolmusz (on behalf of all the authors)